# Magnetic Resonance Imaging Segmentation via Weighted Level Set Model Based on Local Kernel Metric and Spatial Constraint

**DOI:** 10.3390/e23091196

**Published:** 2021-09-10

**Authors:** Jianhua Song, Zhe Zhang

**Affiliations:** 1College of Physics and Information Engineering, Minnan Normal University, Zhangzhou 363000, China; 2Electronic Engineering College, Heilongjiang University, Harbin 150080, China; 2171313@s.hlju.edu.cn

**Keywords:** image segmentation, intensity inhomogeneity, level set, brain magnetic resonance imaging

## Abstract

Magnetic resonance imaging (MRI) segmentation is a fundamental and significant task since it can guide subsequent clinic diagnosis and treatment. However, images are often corrupted by defects such as low-contrast, noise, intensity inhomogeneity, and so on. Therefore, a weighted level set model (WLSM) is proposed in this study to segment inhomogeneous intensity MRI destroyed by noise and weak boundaries. First, in order to segment the intertwined regions of brain tissue accurately, a weighted neighborhood information measure scheme based on local multi information and kernel function is designed. Then, the membership function of fuzzy c-means clustering is used as the spatial constraint of level set model to overcome the sensitivity of level set to initialization, and the evolution of level set function can be adaptively changed according to different tissue information. Finally, the distance regularization term in level set function is replaced by a double potential function to ensure the stability of the energy function in the evolution process. Both real and synthetic MRI images can show the effectiveness and performance of WLSM. In addition, compared with several state-of-the-art models, segmentation accuracy and Jaccard similarity coefficient obtained by WLSM are increased by 0.0586, 0.0362 and 0.1087, 0.0703, respectively.

## 1. Introduction

Magnetic resonance imaging (MRI) is a part and parcel of medical imaging for its virtues such as rapid acquirement, non-intrusive and painless [1]. MRI images have been widely used in treatment evaluation, brain development monitoring, diagnosis, and so on [2,3,4,5]. However, the quality of MRI images is often influenced by various reasons such as low-contrast, noise and intensity inhomogeneity (IIH) during the imaging process, so it is not convenient to accurately segment and extract brain tissues. In addition, intensity inhomogeneity is called as bias field due to the property of slowly varying pixels in the same tissue [6]. Image segmentation has been extensively and deeply studied in computer vision due to its widespread application [7,8,9,10]. In the aspect of biomedical image analysis, it is a fundamental and complex task, which aims at assigning each pixel or voxel to the region with the same anatomical or biological meaning [11,12,13,14]. For example, by segmenting brain tissue, the doctors can detect and judge the changes of brain volume in the physiological or pathological state. Therefore, the need for methods that can accurately segment images has risen rapidly. Many segmentation methods have been presented by researchers to segment MRI images corrupted by intensity inhomogeneity, noise, and so on. These methods can be divided into several mainly categories: boundary-based methods [15,16,17,18,19], threshold-based methods [20,21,22], clustering methods [23,24,25,26,27], region growing methods [28,29], graph cuts methods [30,31,32] and deep learning method [33,34].

Deep learning is applied to medical image processing, segmentation and classification accuracy has been greatly improved, it effectively promotes the development of brain image automatic processing technology. However, this technology also has some problems. Deep learning mostly depends on expert-labeled samples for training. Small-scale sample training has poor results. However, it is extremely difficult, expensive, and time-consuming to obtain large-scale expert-labeled brain segmentation samples. Most of the current studies have focused on the segmentation of glioma and meningioma, and there are few studies on segmentation for other brain lesions. Most deep learning networks are still in the experimental stage, and it will take a long time before clinical application. Because boundary-based methods do not need large-scale expert labeled samples, and the segmentation results have obvious and smooth contour. Therefore, boundary-based methods have been widely applied to MRI image segmentation.

Boundary-based methods depend on the intensity information of MRI image to detect the boundary of interested region [35]. Level set model is one of the most frequently used and well-established boundary-based methods for image segmentation and has attracted significant attention in the past few decades. It regards the contour called level set function as zero level set of higher dimensional function, which can transform the movement of contour into the evolution of level set function implicitly. Level set models can be subdivided into edge-based models and region-based models again [36].

Gradient information is employed by edge-based model as a constraint condition to detect the boundary of an interesting region. This method is particularly effective for the image with clear boundaries. However, for the image with weak boundaries, the level set function is usually difficult to converge to the correct boundaries due to the destruction of noise and intensity inhomogeneity [37]. Li et al. designed a distance regularized level set evolution (DRLSE) scheme [38], in which the image gradient information is used as driving force and regularization term is introduced into energy function to adjust the deviation between signed distance function and level set function to eliminate the defect of continuous re-initialization. Nevertheless, DRLSE only uses the image gradient information as a constraint condition to extract the boundary of each tissue. As a result, DRLSE is helpless to image with more than one target or weak boundaries. Then, Zhang et al. presented a level set evolution method driven by enhanced term to improve the effectiveness of DRLSE [39], the optimized area energy term was defined in their study to detect the boundaries of an image with several disjoint targets. Although the accuracy of this model is improved effectively, it is still sensitive to noisy pixels and initial contour.

Region-based models rely on the similarity of pixels belonging to the same anatomical tissue and relevant statistical information as the constraint conditions to detect the boundary of an interesting region. Region-based models have more advantages than edge-based models since they use region information rather than gradient information to drive the motion of level set function [40]. Therefore, they can be used to segment the images with weak boundaries and low contrast, and it is often insensitive to the position of the initial contour. The most representative model is Chan-Vese (CV) model [41]. It is not only suitable for the case of piecewise smooth but also for the case of piecewise constant. However, it presupposes that the intensity of each tissue is uniform, which means it cannot deal with the inhomogeneous MRI images well. Later, Li et al. a local clustering criterion in local intensity clustering (LIC) to process inhomogeneous intensity images and estimate bias field simultaneously [42]. LIC uses local intensity information and kernel function to homogenize the intensity of each tissue, which can effectively improve the accuracy of segmentation results. However, due to lack of spatial constraints, LIC is sensitive to noise, and the estimated bias field cannot satisfy the slowly varying property. Then, Feng et al. presented a local inhomogeneous intensity clustering (LINC) to improve the segmentation performance of LIC and ensure the properties of bias field [43]. More specifically, LINC is a LIC model with multiplicative intrinsic component optimization (MICO) [44], which utilizes clustering criterion defined in LIC and the basis function of bias field employed in MICO to estimate bias field. MICO employed the fourth-order Legendre polynomials as the basis function to model bias field. However, from the experimental results, LINC is still sensitive to noise, especially for images with weak boundaries. Recently, local information attracts more attention than global information since it can directly reflect the similar property of the central pixel according to its neighbor pixel. Zhou et al. proposed a correntropy-based level set method (CLSM) for correcting bias field and segmenting medical images [45]. CLSM employed both local intensity information and bias field information to define a local bias-field-corrected image fitting energy, which is effective in segmenting inhomogeneous intensity medical images. Further, an optimal segmentation scheme was defined by Huang et al. using a fast level set model [46]. This model provided an optimal partition for every pixel in the entire image domain to inhomogeneous intensity images. In order to eliminate the influence of the bias field, MICO uses a fourth-order Legendre polynomial as the basis function to model bias field. In the process of energy minimization, it is a good modeling idea to find the best coefficient of the basis function to ensure the property of bias field. However, the usage of a fourth-order polynomial in its basis function will lead to higher computational complexity, and it is necessary to simplify the computational efficiency. Finally, the advantages and disadvantages of the relevant studied methods are summarized in Table 1.

Although many improvement measures have been proposed for the level set-based image segmentation scheme, there are still some shortcomings in the initialization of the level set function, the stability and accuracy of model evolution. Further, not only the intensity information but also more information such as variance, spatial position and gray-level should be incorporated into energy function to deal with MRI images. In this study, a weighted level set model (WLSM) based on local kernel metric and spatial constraint is proposed to segment brain MRI images corrupted by noise and intensity inhomogeneity. A neighborhood weighting method consisting of local variation, spatial distance and gray-level information is constructed to improve the resolution of low-contrast brain MRI images and reduce noise interference. Then, the coefficient of data term in energy function is enhanced by fusing fuzzy spatial constraint, which is more suitable for inhomogeneous MRI image segmentation. In addition, in order to ensure the stability of the level set function in the evolution process of level set, the single potential function used in the distance regularization term of the energy function is improved to a new double-well potential function.

The organization of the rest of this study is as follows. Section 2 describes the WLSM proposed in this study in detail. Section 3 presents the experimental results consisting of visual experiments and quantitative evaluation. Finally, the conclusion of this study is given in Section 4.

## 2. Weighted Level Set Model

Brain MRI images are easily corrupted by noise and bias field, which makes it difficult for many existing algorithms to obtain satisfactory segmentation results. To this end, in response to practical problems, a weighted level set model combining local multiple-information is proposed, which ensures accurate segmentation of brain MRI images by introducing kernel metrics and fuzzy spatial membership constraints.

### 2.1. Weighted Neighborhood Information

The improvement of image quality in this study mainly relies on the neighborhood information of the central pixel *x_i_* especially effective for the low-contrast brain MRI images, where white matter (WM) and gray matter (GM) are intertwined with each other and the boundary is blurred. Local variation, spatial distance and gray-level difference are used simultaneously to construct the weighted neighborhood information, which will be described separately in this subsection.

#### 2.1.1. Local Variation Coefficient

The local variation depends on the intensity mean and variance of pixels fall into the local window of central pixel *x_i_*. Variance, which is defined as the degree of deviation between pixels and mean, can reflect the difference between pixels intuitively. Therefore, the local variation coefficient *C_ij_* of each neighbor pixel *x_j_* for the center pixel *x_i_* is defined as:(1)$$C_{ij}=\frac{var\left(x_{j^{\prime }}\right)}{{\left({\bar{x}}_{j^{\prime }}\right)}^{2}},\;j^{\prime }\in N_{j},$$
where $x_{j^{\prime }}$ denotes the gray scale of neighborhood pixel of *x_j_*, that is, *x_j_* is the neighbor pixel of *x_i_*, while $x_{j^{\prime }}$ is the neighbor pixel of *x_j_*. *var*(*x*) and $\bar{x}$ stand for the variance and mean of the corresponding image block, respectively. $N_{j^{\prime }}$ denotes the neighborhood centered on $x_{j^{\prime }}$. The greater the difference between pixels $x_{j^{\prime }}$, the larger the value of *C_ij_*, which means that it reflects the degree of homogeneity of pixels $x_{j^{\prime }}$. If $x_{j^{\prime }}$ is similar to *x_j_*, such as the region with homogeneous intensity, the value of *C_ij_* will be small, and vice versa. Thus, it can utilize more contextual information to explore the local variation because it is computed in the neighborhood of *x_j_*. Then, *C_ij_* is projected into the kernel metric space, and its larger value will result in a smaller gray level weight because of the fast damping property of kernel function. Both Gaussian kernel function and Laplace kernel function have the ability to restrain noise, but the performance of Gaussian kernel function depends largely on the choice of coefficient. Laplace kernel function is the variant of Gaussian kernel function, and it is insensitive to the choice of coefficient. Therefore, the weighting coefficient *γ_ij_* of each pixel *x_j_* in the neighborhood of *x_i_* using Laplace kernel can be expressed as:(2)$$\gamma_{ij}=exp\left(-\frac{C_{ij}}{l}\right),\;j\in N_{i},$$
where *l* is a bandwidth constant. *N_i_* is the neighborhood centered on *x_i_*.

#### 2.1.2. Adaptive Spatial Measure

The local spatial measure used in some typical studies is still the Euclidean distance due to its computationally simple. However, Euclidean distance will easily result in incorrect measure results for the images destroyed by noise, intensity inhomogeneity, and so on. Therefore, a nonlinear version of the linear distance is constructed using kernel metric to improve Euclidean distance, which has been used in fuzzy spatial C-means [47] and support vector clustering [48]. Gaussian kernel function is an efficient distance measure, which is given by:(3)$$D_{ij}^{2}=exp\left(-\frac{{\left\|x_{j}-x_{i}\right\|}^{2}}{\sigma }\right),\;j\in N_{i},$$
where *σ* is the bandwidth coefficient that is often set to a constant in most studies. According to the above description about Gaussian kernel function, it is sensitive to the choice of coefficient and *σ* has a significant effect on its performance, so *σ* should not be fixed as a constant. Thus, WLSM enhances the Gaussian kernel function by calculating *σ* based on the average variance of the distance between *x_j_* and *x_i_* to adaptively update the value of *σ*. Let Euclidean distance between *x_j_* and *x_i_* is:(4)$$d_{ij}={\left\|x_{j}-x_{i}\right\|}^{2},\;j\in N_{i}.$$

The degree of deviation between *x_j_* and *x_i_* can be seen from *d_ij_*. Then, the mean distance $\bar{d}$ of *d_ij_* is given by:(5)$$\bar{d}=\frac{\sum_{j=1}^{n}{\left\|x_{j}-x_{i}\right\|}^{2}}{n},\;j\in N_{i},$$
where *n* is the number of *x_j_*. Accordingly, *σ_i_* can be calculated based on mean square variance of these distances
(6)$$\sigma_{i}={\left(\frac{\sum_{j=1}^{n}{\left\|d_{\mathrm{ij}}-\bar{d}\right\|}^{2}}{n}\right)}^{\frac{1}{2}},\;j\in N_{i}.$$

Finally, the enhanced Gaussian kernel distance is rewritten as:(7)$$D_{ij}^{2}=exp\left(-\frac{{\left\|x_{j}-x_{i}\right\|}^{2}}{\sigma_{i}}\right),\;j\in N_{i}.$$

Comparing Equations (3) and (7), the constant *σ* is replaced by the variable *σ_i_*. In order to visually display the role of $D_{ij}^{}$, take an eight-neighborhood image patch in the white matter region of brain MRI image, and the value of $D_{ij}^{}$ after normalization is shown in Figure 1. There are two pixels in the image patch are destroyed in Figure 1a, and the weights of noisy pixel and the pixel with the same intensity obtained from enhanced Gaussian kernel distance in Figure 1b are greatly different. This is partly because the damping degree of kernel distance will decrease the weight of noisy pixels, but mainly because the bandwidth coefficient *σ_i_* of the enhanced Gaussian kernel distance can be calculated adaptively on the basis of the mean square variance of the distance between neighbor pixels and the central pixel.

#### 2.1.3. Synthetic Weight of Neighborhood Term

The local gray level information *x_j_* with gray level weight *γ_ij_* and Gaussian kernel function $D_{ij}^{2}$ can be used to update the current central pixel *x_i_*.
(8)$$x_{i}=\sum_{j=1}^{N}\gamma_{ij}\cdot D_{ij}^{2}\cdot x_{j},\;j\in N_{i}.$$

An original image with 5% noise obtained from BrainWeb [49] and its corresponding enhanced image using weighted neighborhood information are compared in Figure 2, and four partial enlarged areas are also displayed. Due to the noise in the original image, the edges of WM and GM are intertwined, it is difficult to distinguish the edge of different brain tissue. However, the enhanced image is much more noise free than the original image. Further, the partially enlarged region can show more details of the boundaries of WM and GM. Therefore, WLSM can improve the quality of images effectively and reduce the influence of noisy pixels.

### 2.2. The Improved External Energy Function

An image *I*(*x*) is divided into *N* disjoint regions *Ω*_1_, …, *Ω_N_*, and the local intensity clustering defined in our previous study [35] described that pixels *x* around *y* with radius *ρ* is given by *O_y_* = {*x*: |*x* − *y*| ≤ *ρ*}. Thus, all pixels in the neighborhood *O_y_* can be divided into *k* classes, which is given by:(9)$$I_{y}^{k}=\{I(x):\;x\in \Omega_{k}\cap O_{y}\}.$$

In other words, this classification criterion means that intensities fall into *O_y_* can be classified into *N* clusters. Subsequently, the clustering function defined based on such criterion is also suitable for this study to classify intensities. According to Ref. [38], the date term *ε* constituting the external energy function of the level set can be expressed as:(10)$$\varepsilon =\int \left(\sum_{k=1}^{N}\int K(y-x){\left|I(x)-w^{T}G(y)c_{k}\right|}^{2}dx\right)dy,$$
where *K*(*y* − *x*) is a function used to control the range of *O_y_*, which has been defined in LIC and should satisfy ∫K(s)ds=1. *c_k_* is the *k*-th clustering center. ***w***^T^*G*(*y*) denotes the bias field *b*(*x*), where *G*(*y*) is the combination of Legendre polynomial functions that can ensure *b*(*x*) is slowly changing and vector ***w*** = (*ω*_1_, …, *ω_N_*)*^T^* is the optimal coefficient. *I*(*x*) denotes the original image, but the weighted neighborhood information can improve the quality of image before the energy function is minimized, so the clustering function can be re-represented as:(11)$$\varepsilon =\int \left(\sum_{k=1}^{N}\int K(y-x){\left|\tilde{I}(x)-w^{T}G(y)c_{k}\right|}^{2}dx\right)dy,$$
where $\tilde{I}$(*x*) denotes the enhanced image of *I*(*x*) using Equation (8).

In Ref. [39], a level set function *ϕ*(*x*) can represent two disjoint regions *Ω*_1_ and *Ω*_2_ by corresponding membership functions *M*_1_(*ϕ*) = *H*(*ϕ*) and *M*_2_(*ϕ*) = 1 − *H*(*ϕ*), where *H*(*ϕ*) is Heaviside function defined by:(12)$$H\left(\phi \right)=\frac{1}{2}\left[1+2\arctan\left(\phi \right)\right].$$

A multi-phase level set function *Φ* = (*ϕ*_1_, …, *ϕ_i_*) can denote *N* disjoint regions *Ω*_1_, …, *Ω_N_* by corresponding membership functions *M_k_*(*Φ*). For example, *ϕ*_1_(*x*) and *ϕ*_2_(*x*) can define three membership functions *M*_1_(*ϕ*_1_, *ϕ*_2_) = 1 − *H*(*ϕ*_1_), *M*_2_(*ϕ*_1_, *ϕ*_2_) = *H*(*ϕ*_1_) ∗ *H*(*ϕ*_2_), and *M*_3_(*ϕ*_1_, *ϕ*_2_) = *H*(*ϕ*_1_) ∗ (1 − *H*(*ϕ*_2_)). Therefore, clustering Function (11) can be converted to the data term of the level set model as follows:(13)$$\varepsilon =\int \left(\sum_{k=1}^{N}\int K(y-x){\left|\tilde{I}(x)-w^{T}G(y)c_{k}\right|}^{2}M_{k}(\Phi )\mathrm{dx}\right)\mathrm{dy}.$$

Clustering center *c_k_* is replaced by the vector ***c*** = (*c*_1_, …, *c_N_*) for convenience. Therefore, the data term *ε* can be re-represented as a new expression *ε*(*Φ*, ***w***, ***c***) about level set function *Φ*, optimal coefficient ***w*** and clustering center vector ***c***. Swapping the order of integrations will not change the data term, so it is written as:(14)$$\varepsilon \left(\Phi ,\;w,\;c\right)=\int \sum_{k=1}^{N}\lambda_{k}e_{k}\left(x\right)M_{k}(\Phi )\mathrm{dx},$$
where *λ_k_* is the weighted coefficient and *e_k_* is expressed as:(15)$$e_{k}\left(x\right)=\int K\left(y-x\right){\left|\tilde{I}(x)-w^{T}G(y)c_{k}\right|}^{2}dy.$$

Its numerical expression is given by:(16)$$e_{k}\left(x\right)=\tilde{I}\left(x\right)1_{K}-2c_{k}\tilde{I}\left(x\right)\left(w^{T}G*K\right)+c_{k}^{2}\left({\left(w^{T}G\right)}^{2}*K\right),$$
where **1***_K_* denotes ∫K(y−x)dy. The weighted coefficient *λ_k_* of data term is often set as *λ*_1_ = … = *λ_k_* = 1 in some typical level set models, such as LIC and LINC. Nevertheless, without any constraints, *Φ* cannot reach or directly pass through the weak boundaries corrupted by low-contrast and thus leads to the interesting region which cannot be completely extracted. Therefore, *λ_k_* should not be set as a global constant at least in the vicinity of zero level set. According to AFLSM [30], *λ_k_* has two significant roles, one is the direction of *Φ* depends on the value of *λ_k_*, the other is to control the speed of *Φ*. In other words, *Φ* should be accelerated when it is far away from the boundaries of the interested region and vice versa. Therefore, *λ_k_* should be improved via contextual information. Fuzzy spatial constraint obtained by fuzzy clustering is suitable for improving *λ_k_* because it assumes pixels in the vicinity of boundaries belong to different clusters according to their membership matrix *u_ki_*, where *k* denotes the *k*-th clustering center and *i* is the *i*-th pixel. Then, the modified weighted coefficient *λ_k_* constrained by *u_ki_* is defined as:(17)$$\lambda_{k}=1-\frac{u_{\mathrm{ki}}}{2\sum_{k=1}^{N}u_{ki}},\;\lambda_{k}\in \left(0,1\right].$$

The difference between the *λ_k_* defined in this study and the one defined in our previous study AFLSM is mainly reflected in two aspects. For one thing, the coefficient *α* of *λ_k_* defined in AFLSM needs to be fine-tuned based on images with different characteristics, but the *λ_k_* defined in WLSM completely depends on the membership matrix *u_ki_*, which means the modified *λ_k_* in this study is fully adaptive and does not need to be adjusted manually. For another, all *λ_k_* defined in WLSM are normalized to ensure the stability of numerical calculation and control the variation of level set function exactly. Thus, the modified *λ_k_* is superior to the one defined in AFLSM and the segmentation accuracy of WLSM can be further improved for the images with weak boundaries.

### 2.3. Internal Energy Function

Li et al. regularized the level set function in LBF [50] to a smooth zero level set by computing the length of its contour, which is defined as the following length term:(18)L(Φ)=∫|∇H(Φ)|dx,
where ∇ is the gradient operator. A conventional level set function has to be reinitialized during each iteration, which is a time-consuming process, especially for a model with thousands of iterations. Accordingly, LBF presented a distance regularization term *P*(*Φ*) that utilizes the unique property |∇*Φ*| = 1 of signed distance functions to avoid the expensive re-initialization during the process of evolution. That is, the distance regularization term can drive *Φ* to approach the signed distance function to eliminate the deviation, which is defined by:(19)P(Φ)=∫p(|∇Φ|)dx,
where *p*(*s*) is a potential function. *p*(*s*) is often set as *p*(*s*) = (*s* − 1)^2^/2, where *s* = 1 is its minimum point used to maintain the property of signed distance functions. However, the single potential function is not always effective in practical applications because of its forward-and-backward diffusion and thus cannot steadily maintain the property |∇*Φ*| = 1 [33]. Therefore, WLSM improved the single potential function by introducing two minimum points *s* = 0 and *s* = 1, which is aimed to solve the drawbacks of LBF and maintain the property |∇*Φ*| = 1 only in a vicinity of zero level set contour. Such a potential function is called the double-well potential function *p*_2_(*s*) because of its two minimum points, which should satisfy the following four conditions:
(1)Two minimum points of double potential function *p*_2_(*s*) should be at *s* = 0 and *s* = 1, respectively;(2)*p*_2_(*s*) is second-order derivable in [0,∞);(3)Function *d_p_*(*s*) defined by *p′*(*s*)/*s* should satisfy |*d_p_*(*s*)| < 1, *s*∈(0,∞);(4)$\underset{s\to 0}{\lim}d_{p}\left(s\right)=\underset{s\to \infty }{\lim}d_{p}\left(s\right)=1.$

Based on the above four constraints, *p*_2_(*s*) is defined in WLSM can be expressed as:(20)$$p_{2}\left(s\right)=\left\{ \begin{array}{l} -\frac{\cos2\pi s}{4\pi^{2}}+\frac{1}{4\pi^{2}},\;s\in \left[0,1\right]\\ (s+1)e^{1-s}+\frac{s^{2}}{2}-\frac{5}{2},\;s\in \left[1,\infty \right) \end{array}\right..$$

The curve of *p*_2_(*s*) and *d_p_*(*s*) are shown in Figure 3. When *p*_2_(*s*) = 0, it can be seen from Figure 3a that *p*_2_(*s*) have two minimum points (no other minimum points) at *s* = 0 and *s* = 1, respectively. The first derivative *p*_2_′(*s*) of *p*_2_(*s*) is given by:(21)$$p_{2}^{\prime }\left(s\right)=\left\{ \begin{array}{ll} \frac{\sin2\pi s}{2\pi }, & s\in \left[0,1\right]\\ -se^{1-s}+s, & s\in \left[1,\infty \right) \end{array}\right..$$

The second derivative *p*_2_*″*(*s*) of *p*_2_(*s*) is given by:(22)$$p_{2}^{\prime \prime }\left(s\right)=\left\{ \begin{array}{l} \cos2\pi s,\;s\in \left[0,1\right]\\ \left(s-1\right)e^{1-s}+1,\;s\in \left[1,\infty \right) \end{array}\right.,$$
which means the novel *p*_2_(*s*) is second-order derivable in [0,∞). Next, the function *d_p_*(*s*) based on *p*_2_′(*s*) is defined as:(23)$$d_{p}\left(s\right)=\frac{p_{2}^{\prime }\left(s\right)}{s}=\left\{ \begin{array}{ll} \frac{\sin2\pi s}{2\pi s}, & s\in \left[0,1\right]\\ -e^{1-s}+1, & s\in \left[1,\infty \right) \end{array}\right..$$

As shown in Figure 3b, it is not difficult to prove that *d_p_*(*s*) derived from *p*_2_(*s*) also satisfies the above conditions Equations (3) and (4), which are to ensure the diffusion rate of *p*_2_(*s*) is bounded. Further, the novel double-well potential function *p*_2_(*s*) not only can eliminate the drawbacks of single potential function but also can penalize its deviation from the signed distance functions. Therefore, WLSM with double potential function *p*_2_(*s*) is more suitable for segmenting images with noise and intensity inhomogeneity.

### 2.4. Energy Formulation and Its Minimization

According to the external function and the internal function, the entire energy formulation can be expressed as:(24)F(Φ, w, c)=ε(Φ, w, c)+νL(Φ)+μP(Φ),
where *ε*(*Φ*, ***w***, ***c***) is the data term defined in Equation (14), *ν* and *μ* are the weighted coefficients of length term *L*(*Φ*) defined in Equation (18) and distance regularization term *P*(*Φ*) defined in Equation (19), respectively.

The method to minimize *Φ* is to solve the following gradient descent flow equation
(25)$$\frac{\partial \Phi }{\partial t}=-\frac{\partial F}{\partial \Phi },\;\Phi =\left(\phi_{1},\cdots ,\phi_{i}\right),$$
where ∂*F*/∂*Φ* is Gâteaux derivative and *t* is a time variable. The right-hand side of the equation shows the steepest descent direction of the energy function *F*. Accordingly, the minimization of *Φ* is given by:(26)$$\frac{\partial \phi_{i}}{\partial t}=-\sum_{k=1}^{N}\frac{\partial M_{k}\left(\phi_{i}\right)}{\partial \phi_{i}}\lambda_{k}e_{k}-\nu \frac{\partial L}{\partial \phi_{i}}-\mu \frac{\partial P}{\partial \phi_{i}},\;\phi_{i}\in \Phi$$
where *e_k_* has been given in Equation (16). The Gâteaux derivative of length term *L*(*Φ*) is given by:(27)$$\frac{\partial L}{\partial \phi_{i}}=-\delta \left(\phi_{i}\right)\mathrm{div}\left(\frac{\nabla \phi_{i}}{\left|\nabla \phi_{i}\right|}\right),\;\phi_{i}\in \Phi$$
where *δ*(*ϕ*) is the derivative of *H*(*ϕ*) and *div*(⋅) is divergence operator. Then, the Gâteaux derivative of distance regularization term *P*(*Φ*) is given by:(28)$$\frac{\partial P}{\partial f_{i}}=-\mathrm{div}\left(d_{p}\left(\left|\nabla \phi_{i}\right|\right)\nabla \phi_{i}\right),\;\phi_{i}\in \Phi$$
where the function *d_p_*(*s*) has been defined in Equation (23).

The method to minimize optimal coefficient ***w*** is to solve the following equation:(29)$$\frac{\partial F}{\partial w}=-2v+2Aw,$$
where ***v*** is a column vector defined by:(30)$$v=\int \left(K*\left(\tilde{I}\left(x\right)\sum_{k=1}^{N}\lambda_{k}c_{k}M_{k}\left(\Phi \right)\right)\right)G\left(y\right)dy,$$
and ***A*** is a matrix defined by:(31)$$A=\int \left(K*\left(\sum_{k=1}^{N}\lambda_{k}c_{k}^{2}M_{k}\left(\Phi \right)\right)\right)G\left(y\right)G^{T}\left(y\right)dy.$$

Thus, according to the expression of ***v*** and ***A***, the minimization of ***w*** is given by:(32)$$w=vA^{-1}.$$

The minimization of *c_k_* is given by:(33)$$c_{k}=\frac{\int \tilde{I}\left(x\right)M_{k}\left(\Phi \right)\left(K*\left(w^{T}G\right)\right)dx}{\int M_{k}\left(\Phi \right)\left(K*{\left(w^{T}G\right)}^{2}\right)dx}.$$

The implementation process of WLSM can be expressed as Algorithm 1.
**Algorithm 1:** WLSM.**Begin****Input:**  original image;  weighted coefficients *ν* and *μ*;**Initialization:**
  bias field *b* and clustering center ***c*** randomly**Process:**
  update the current central pixel *x**_i_* for all pixels according to Equation (8);  while |***c***^(*n*)^ − ***c***^(*n*−1)^| > 0.001    update level set function *ϕ* according to Equation (25);    update bias field *b* according to Equation (32);    update clustering center *c* according to Equation (33)**Output:**  enhanced image; corrected image; segmentation result; estimated bias field**End**

## 3. Experiments and Results

In this section, WLSM is used to segment synthetic and real MRI images and compared with state-of-the-art models. Firstly, it is used to segment MRI images with varying degrees of noise to demonstrate that the weighted neighborhood information embedded in WLSM can enhance the quality of MRI images and improve segmentation accuracy (SA). Then, it is used to correct inhomogeneous intensity MRI images and estimate bias field. Finally, the sagittal, coronal and axial slices of synthetic and real MRI images are segmented by WLSM to show its performance from different aspects. BrainWeb is a simulated brain database (SBD) and includes a set of data generated by an MRI simulator that is close to real brain MR images [49], and it was developed by the brain imaging center of the Montreal Neurology Institute at McGill University. The brain MR image data in SBD is composed of a three-dimensional matrix of 181 × 217 × 181 voxels, which can simulate T1, T2 and PD (proton-density) weighted brain MRI images, and the slice thickness, noise level and intensity inhomogeneity can be set by oneself. According to the requirement of the research task, 3D images can be sliced and extracted from three planes (sagittal, coronal and axial) to obtain 2D image data. The real images used in this study are from the internet brain segmentation repository (IBSR) database provided by Formal Measurement Center (CMA) of Massachusetts General Hospital [51]. IBSR database is a set of clinical data generated by real MRI scans, which contains different levels of noise, and there are also varying degrees of intensity inhomogeneity, covering various problems that may arise in real MR data segmentation. Unless otherwise specified, parameters used in WLSM are fixed as *ν* = 0.001 × 255 × 255, *µ* = 1, and ∆*t* = 0.1. All experimental results were implemented by Matlab R2019a on a computer with Intel Core i5-8300H 2.3 GHz CPU, 8 GB RAM, Windows 10 operating system, and the segmentation results with final zero-level contours of *ϕ*_1_ in red and *ϕ*_2_ in blue are shown.

### 3.1. Results on Noisy Images

First of all, WLSM model is applied to 20 T1-weighted normal brain MRI images with 181 × 217 pixels, 1 mm slice thickness, 5% noise, without intensity inhomogeneity, their cranium and blood vessels have been dislodged before segmentation processing. In Figure 4, serial numbers of the sliced images from the axial plane are 75 and 85, respectively. Original images in the first column are destroyed by noise especially in the regions that WM and GM are interweaved with each other, which can be easily seen from the partial enlarged regions marked by the green rectangles. For comparison, the enhanced images using weighted neighborhood information in the second column are more noise free than original images and the boundary of each tissue can be clearly distinguished because of the constraints of weighted neighborhood information. To be specific, WLSM utilizes the neighbor information consisting of local variation, spatial distance and gray-level information to update the current central pixel. The neighbor pixels in the same tissue will occupy bigger weights and vice versa. Therefore, pixels located in the weak boundaries can be categorized into the corresponding tissue according to their weighted neighborhood information. Subsequently, benefitting from such the weighted neighborhood information, level set functions are able to accurately extract the boundary of brain tissue, which can be observed from the third column, the last column is the ground truth.

The higher the noise level, the more difficult it is to accurately segment the noisy MRI images. Thus, brain MRI images with the difference of noise level are segmented by WLSM to testify its robustness to noise. The sliced images with 3%, 5%, 7% and 9% noise are displayed in the first to the fourth columns of Figure 5, respectively. The quality of original images in the first row gradually decreases with the increase of noise, which will easily result in non-robust segmentation results. However, the segmentation results obtained by WLSM in the second row have no obvious discrepancy even in the case of severe noise, which can show its sturdy robustness to the different noise levels.

Because fuzzy C-means (FCM) clustering and its variants are often used to segment brain MRI images, and in the WLSM model, the spatial neighborhood constraint is also realized by the idea of the FCM. Therefore, to further compare the segmentation performance, the two clustering algorithms are also compared. One is the standard FCM algorithm, and the other is the adaptively regularized kernel-based fuzzy C-means clustering (ARKFCM) algorithm in Ref. [52]. Figure 6 shows the comparison results of FCM, LINC [38], MICO [39], ARKFCM [52] LIC [37] and WLSM on brain MRI images with 5% noise. FCM is very sensitive to noise, and the segmentation effect is poor. LINC employs the local clustering function to define the energy function, which can cluster inhomogeneous intensity in the neighborhood. However, all pixels including noisy pixels are clustered in the local clustering function, so the performance of LINC is easily influenced by noise. Accordingly, it can be seen from Figure 6c that LINC obtained incorrect results in the face of images corrupted by noise. MICO is based on the hard clustering to segment images, where hard clustering assumes that each pixel in the entire image domain only belongs to one cluster, so it is also sensitive to noise. Thus, the segmentation results obtained by MICO in Figure 6d are easily influenced by noise pixels. In addition, the results have numerous wrong contours representing noisy pixels. ARKFCM employs the heterogeneity of grayscales in the neighborhood and exploit this measure for local contextual information and replace the standard Euclidean distance with Gaussian radial basis kernel functions, which can effectively suppress the noise in the image, but ARKFCM cannot estimate the bias field in brain MRI images, and it is susceptible to the interference of intensity inhomogeneity, as shown in Figure 6e. LIC uses the intensity clustering property to define a global criterion function, which can estimate bias field and segment images simultaneously. However, it is also sensitive to noise because of the lack of spatial constraints. As shown in Figure 6f, LIC cannot accurately extract the boundary between GM and WM and thus leads to over-segmentation. By contrast, the segmentation results by WLSM displayed in Figure 6g shows superior performance than the other five models.

In order to quantitatively analyze the experimental results, segmentation accuracy (SA) is often used to evaluate the segmentation performance of algorithms, which is defined as:(34)$$SA\left(S,G\right)=\sum_{i=1}^{k}\frac{S_{i}\cap G_{i}}{\sum_{j=1}^{k}G_{j}},$$
where *k* is the number of clusters, *S**_i_* is the pixel number belonging to the *i*th cluster found by the algorithm and *G_i_* is the pixel number belonging to the *i*th cluster in GT. The higher the value of SA means that model can obtain more accurate results on the segmented images. In this experiment, we selected 10 groups of brain MRI images as experimental samples to compare the segmentation results. SA of FCM, LINC, MICO, ARKFCM, LIC and WLSM are shown in Table 2. Both the SA of WM and GM and the average SA can show the superiority of WLSM, which objectively proves that WLSM obtains more accurate segmentation results on noisy MRI images.

### 3.2. Results on Inhomogeneous Intensity Images

In general, MRI images are often corrupted by noise and intensity inhomogeneity simultaneously during the process of imaging, so the ability to correct inhomogeneous intensity and estimating bias field is the part and parcel of the performance of the model. Therefore, in the experiment of this subsection, WLSM is used to segment brain MRI images with 3% noise and varying degrees of intensity inhomogeneity to testify its performance. Images with 3% noise and 60%, 80% and 100% intensity inhomogeneity are displayed on the first row to the third row of Figure 7, respectively. The corrected images in the second column are much more homogeneous than one of the original images in the first column, and the segmentation results in the third column have no obvious discrepancy whether in the case of lower inhomogeneous intensity or higher inhomogeneous intensity, which illustrates that WLSM can correct inhomogeneous intensity and segment images simultaneously without being influenced by noise and intensity inhomogeneity. Benefitting from the third-order orthogonal Legendre functions, the bias field in the fourth column estimated by WLSM satisfies the property of smooth changing. Then, the histograms of original and corrected images are plotted in Figure 8 to objectively compare the images quality. The histograms of original images have no obvious peaks that represent corresponding tissues because of the influence of noise and intensity inhomogeneity. In contrast, there are well-defined peaks in histograms of the corrected images, which objectively reflect the improvement of the image quality.

In this experiment, because FCM and ARKFCM algorithms do not have the ability to estimate the bias field, we only compare the segmentation results of LINC, MICO, LIC and WLSM. In Figure 9, the tested brain MRI images are corrupted by 5% noise and 100% intensity inhomogeneity. Original images, corrected images, segmentation results and bias field obtained by each model are shown in the first to the fourth rows, respectively. As shown in the first and second columns of the second row, the corrected images obtained by LINC and MICO are still inhomogeneous especially in the regions that WM and GM are interweaved with each other. It can be seen from the third column of the last row that LIC cannot ensure the property of the bias field due to a lack of smooth basis functions. The corrected image obtained by WLSM is more homogeneous than the original image and the one obtained by LINC, MICO and LIC, respectively. In addition, the third-order orthogonal Legendre functions used by WLSM can ensure the especial property of bias field. Therefore, WLSM can estimate bias field excellently and the corrected image will much more homogeneous after removing the estimated bias field. Then, the iterations and calculative time of LINC, MICO, LIC and WLSM are contrasted and the results are shown in Table 3. The size of the original image in the first column of Figure 9 is 181 × 217. It can be observed from the last row of Table 3 that WLSM consumes less calculative time than the other three models in segmenting brain MRI images corrupted by both noise and intensity inhomogeneity.

Jaccard similarity coefficient (JSC), the measure of similarity between two sets, is applied to quantitatively compare the performance of segmentation model. The definition of JSC is given by:(35)$$JSC\left(S,\;S_{G}\right)=\frac{\left|S\cap S_{G}\right|}{\left|S\cup S_{G}\right|},$$
where *S* is the segmentation result obtained by each model and *S*_*G*_ is GT. The higher value of JSC indicates that the corresponding model is better than other methods. To quantitatively compare the accuracy of LINC, MICO, LIC and WLSM, the JSC obtained by the above four models is plotted in the box plots of Figure 10. As can be seen, the JSC of WM and GM obtained by WLSM exhibit the higher values among the four models, which can quantitatively demonstrate that WLSM can reach higher segmentation accuracy in comparison with LINC, MICO and LIC.

### 3.3. Results on Sagital, Coronal and Axial Slices of Images

In this subsection, the sagittal, coronal and axial slices of synthetic and real brain MRI images are segmented to show the effectiveness of models. Firstly, the segmentation results of synthetic brain MRI images obtained by FCM, LINC, MICO, ARKFCM, LIC and WLSM are shown in Figure 11. The sliced images from the sagittal, coronal and axial planes are displayed in the first to the third rows of Figure 11, respectively. Original images in the first column are corrupted by noise and intensity inhomogeneity simultaneously. FCM is not robust to noise and bias field, LINC still cannot accurately extract the interested tissues and thus leads to segmentation errors. MICO cannot distinguish the boundary of WM and GM, because their gray values are too similar in the case of severe inhomogeneous intensity, which can be seen from the sagittal and axial slice images in the second and third rows of the fourth column. ARKFCM cannot eliminate the influence of the bias field in the brain MRI images, and it has encountered a problem in the segmentation of white matter, as shown in the fifth column. LIC is sensitive to noise pixels because of the deficiency of spatial constraints. Hence, the segmentation results shown in the sixth column are still corrupted by noise especially in the boundary of WM and GM. As shown in the seventh column, WLSM can accurately extract the boundary of WM and GM whether on the sagittal, coronal or corrupted slice images, which means WLSM is effective on synthetic images with different planes.

Finally, WLSM is applied to segment the sagittal, coronal and axial slices of real brain MRI images and is also compared with FCM, LINC, MICO, ARKFCM and LIC to further demonstrate the effectiveness of WLSM. The segmentation results of real images obtained from the four models are shown in Figure 12. The real sagittal, coronal and axial slices are shown in the first to the third rows of Figure 12, respectively. Compared with all the experimental results, WLSM is more accurate than FCM, LINC, MICO, ARKFCM and LIC, which illustrates that it can be employed to segment real brain MRI images with different planes without being influenced by the skull.

## 4. Discussion

The edge-based level set model is to segment and extract the corresponding brain tissue according to the boundary of the target. Therefore, for images with weak tissue boundaries or low contrast, the conventional level set function cannot accurately identify the boundaries of each tissue. Firstly, the WLSM algorithm targets the weak boundary of low contrast images and proposes a method to improve the edge contrast of brain tissue. This method uses local variance, spatial distance and gray information to update the current central pixel. The updated image has high contrast at the weak boundary of the brain tissues, so it lays a sound foundation for the subsequent level set function to extract the target edge. Secondly, the third-order Legendre polynomial function with orthogonality is used as the basis function to estimate the bias field, and its optimal coefficient is found in the iterative process. Compared with the traditional linear bias field estimation model, WLSM can ensure that bias field variation is smooth and slow. Thirdly, taking the membership function of FCM as the adaptive coefficient of the data item in the energy function changes the defect that the original coefficient is constant. It can not only overcome the problem that level set function is sensitive to initialization but also adaptively control the evolution of level set function. Finally, the single potential function in distance regularization term is replaced by a novel double-well potential function, which can effectively avoid reinitialization and maintain the accuracy and stability of the evolution of the level set model.

Of course, the proposed algorithm still needs to be improved. In the application of image segmentation, WLSM model only segments normal brain MRI images, and does not process brain MRI images with diseases such as multiple sclerosis, brain tumors and further research can be carried out on such images. Furthermore, this paper only studies the segmentation algorithm of 2D MRI slice images and does not directly segment 3D MRI images. In the future, we will further improve WLSN model combining deep learning to segment images in 3D and try to apply the multiphase formulation to segment brain tissues on public image repositories.

## 5. Conclusions

In this study, according to the deficiencies in the initialization of level set function, the stability and accuracy of model evolution in the current segmentation model, a weighted level set model (WLSM) based on local kernel metric and spatial constraints is proposed to segment brain MRI images corrupted by noise and intensity inhomogeneity. WLSM can not only accurately control the evolution of level set function, but also overcome the problem of model reinitialization. Compared with state-of-the-art models, the visual experiments including both synthetic and real MRI images demonstrate the superiority of WLSM.

## Figures and Tables

**Figure 1 entropy-23-01196-f001:**
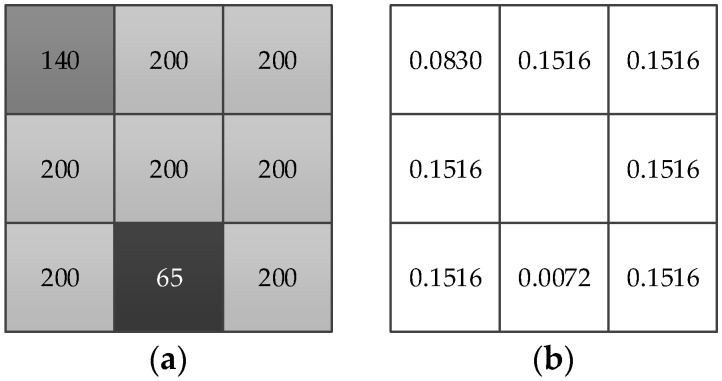
Gray level value and its *D_ij_* value in an image patch. (**a**) Gray level value in an image patch and (**b**) the corresponding *D_ij_* value.

**Figure 2 entropy-23-01196-f002:**
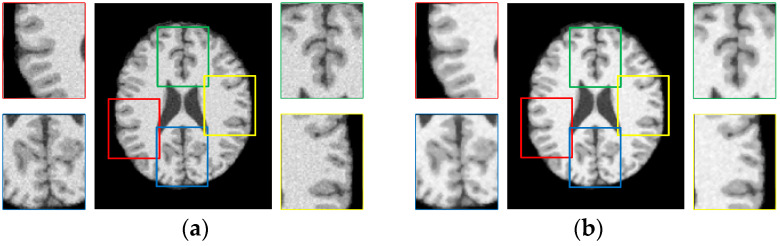
Comparison of noisy image and weighted image on brain MRI. (**a**) Original image and (**b**) the weighted image.

**Figure 3 entropy-23-01196-f003:**
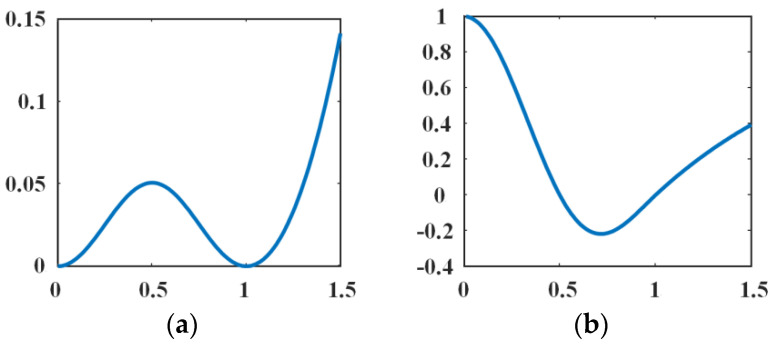
The curve of *p*_2_(*s*) and *d_p_*(*s*). (**a**) The curve of *p*_2_(*s*) and (**b**) the curve of *d_p_*(*s*).

**Figure 4 entropy-23-01196-f004:**
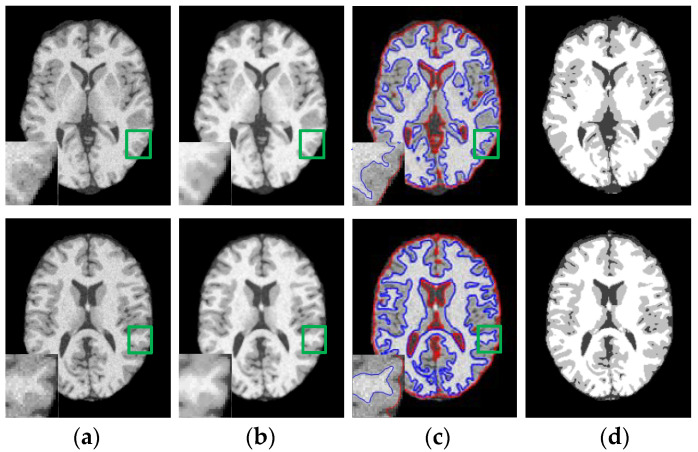
Segmentation results of WLSM on brain MRI images corrupted by 5% noise. (**a**) Original images; (**b**) enhanced images; (**c**) segmentation results and (**d**) ground truth (GT).

**Figure 5 entropy-23-01196-f005:**
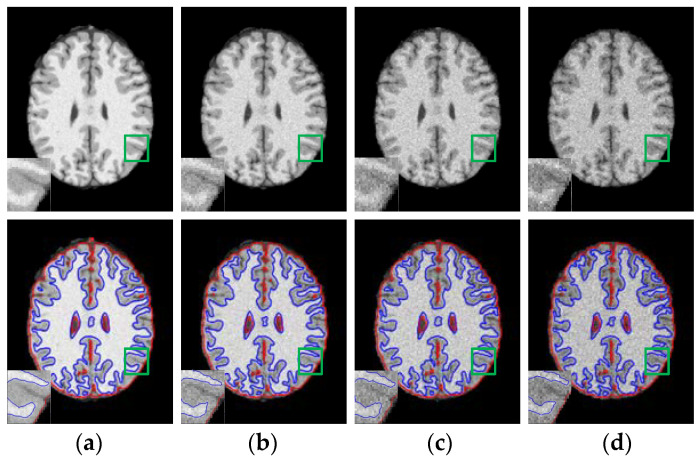
Image segmentation results of different noise level. (**a**) 3% noise; (**b**) 5% noise; (**c**) 7% noise and (**d**) 9% noise.

**Figure 6 entropy-23-01196-f006:**
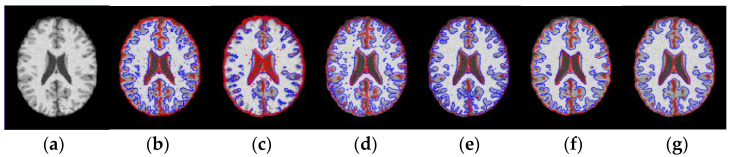
Segmentation results of the six models on brain MRI image corrupted by 5% noise. (**a**) original image; (**b**) result of FCM; (**c**) result of LINC; (**d**) result of MICO; (**e**) result of LIC; (**f**) result of ARKFCM and (**g**) result of WLSM.

**Figure 7 entropy-23-01196-f007:**
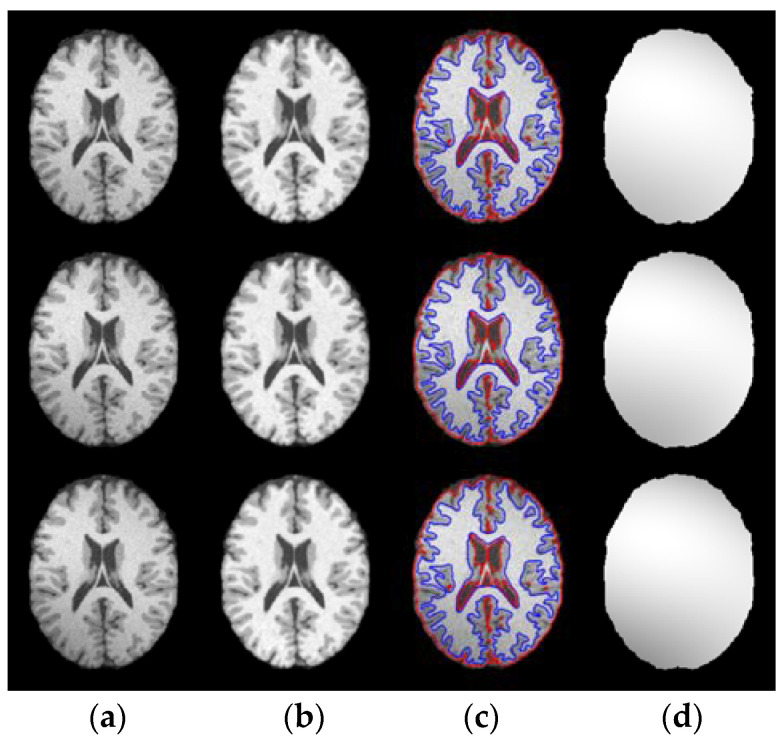
Segmentation results of the proposed model on brain MRI images with different levels of bias field. (**a**) Original images; (**b**) the corrected images; (**c**) segmentation results and (**d**) the estimated bias field.

**Figure 8 entropy-23-01196-f008:**
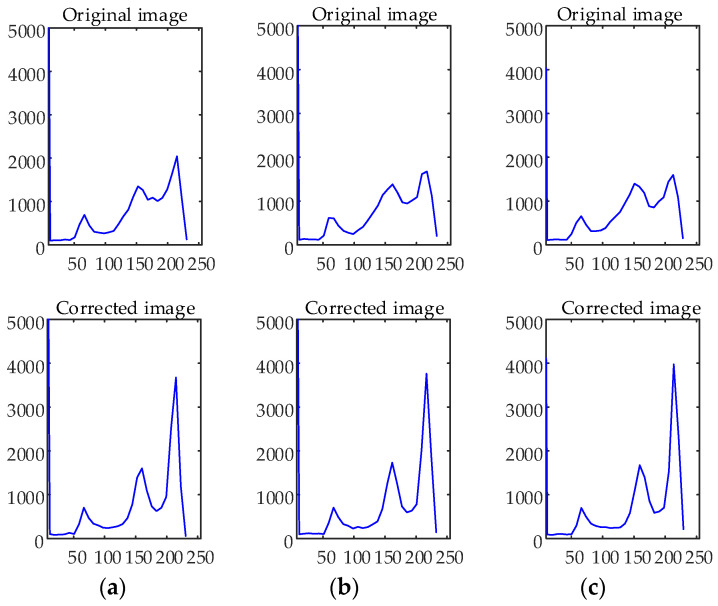
The histograms of original and bias field correction images: (**a**) 60% intensity inhomogeneity; (**b**) 80% intensity inhomogeneity and (**c**) 100% intensity inhomogeneity.

**Figure 9 entropy-23-01196-f009:**
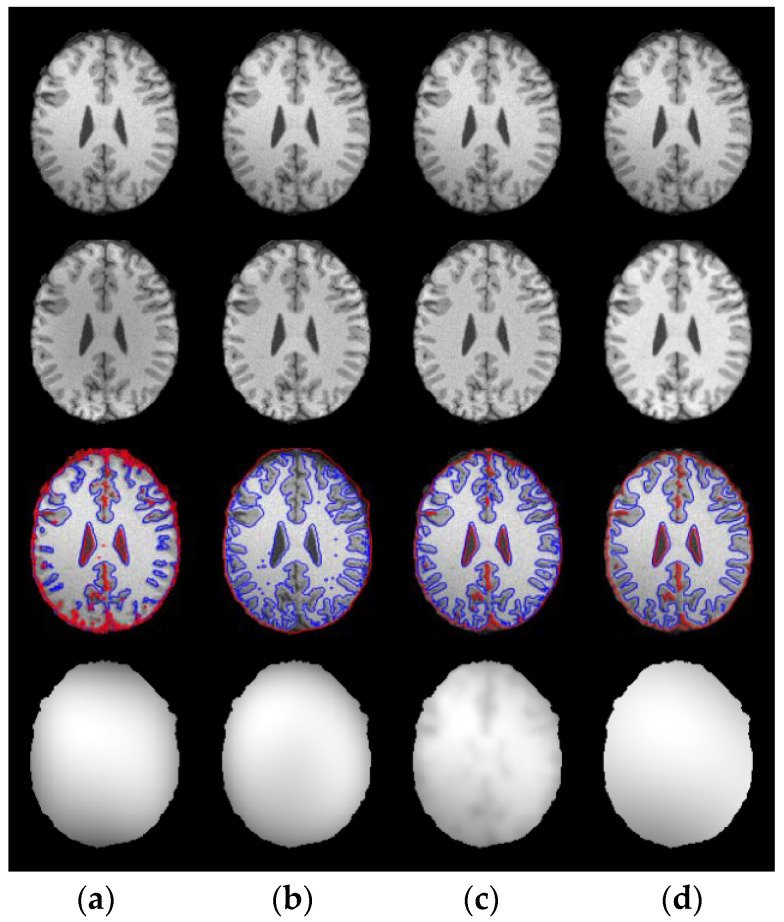
Segmentation results of the four models on image corrupted by 5% noise and 100% intensity inhomogeneity: (**a**) results of LINC, (**b**) results of MICO, (**c**) results of LIC, (**d**) results of the WLSM.

**Figure 10 entropy-23-01196-f010:**
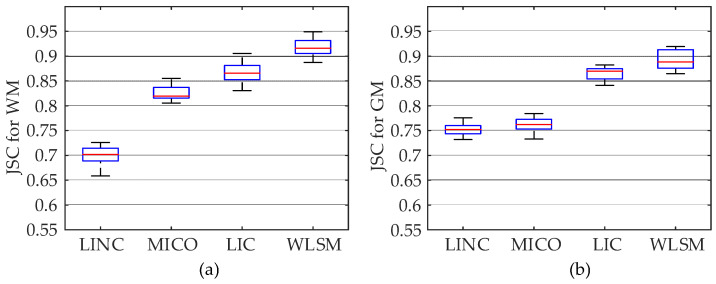
JSC comparison of four models. (**a**) JSC of WM and (**b**) JSC of GM.

**Figure 11 entropy-23-01196-f011:**
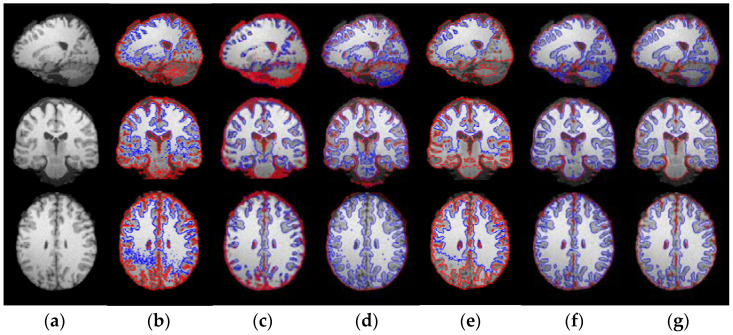
Segmentation comparison of WLSM with FCM, LINC, MICO, ARKFCM and LIC on the sagittal, coronal and axial slices of synthetic brain MRI images: (**a**) original images; (**b**) results of FCM; (**c**) results of LINC; (**d**) results of MICO; (**e**) results of ARKFCM; (**f**) results of LIC and (**g**) results of WLSM.

**Figure 12 entropy-23-01196-f012:**
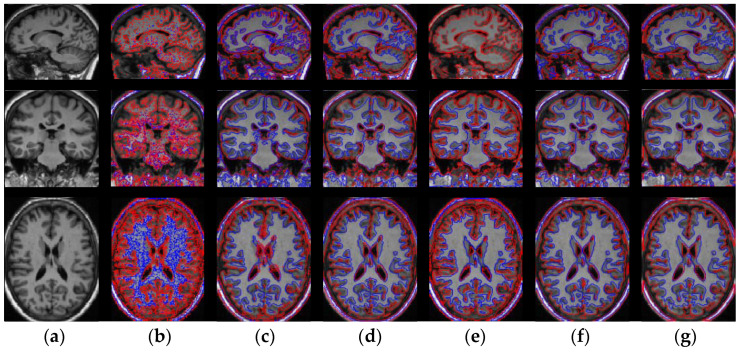
Segmentation results of the six models on the sagittal, coronal and axial slices of real brain MRI image: (**a**) original images; (**b**) results of FCM; (**c**) results of LINC; (**d**) results of MICO; (**e**) results of ARKFCM; (**f**) results of LIC and (**g**) results of WLSM.

**Table 1 entropy-23-01196-t001:** Advantages and disadvantages of the related segmentation methods.

Name of the Method	Advantages	Disadvantages
CV [41]	Able to detect interior contours and thus, could be used for medical images with weak boundaries. Piecewise smooth model could work for medical images with IIH.	Limited by images with complicated background and irregular intensity. Piecewise constant case only works with images having homogeneous regions.
LIC [42]	Able to estimate bias field and segment brain tissues simultaneously.	Bias field model is an idealized model without fully considering its own properties.
MICO [44]	The slowly and the smoothly varying property of the bias field is ensured by a linear combination of a given set of smooth basis functions.	The model is not a level set method and is sensitive to noise without considering local neighborhood information.
LINC [43]	A local clustering criterion function is defined to cluster intensities in the neighborhood for utilizing local neighborhood information.	All pixels including noise pixels are clustered into local clustering criterion, so LINC is sensitive to noise and weak boundaries.
CLSM [45]	Incorporate the correntropy criterion into the energy function of local bias-field-corrected fitting image.	Difficult to discriminate pixels having same or minor differences between foreground and background, program execution efficiency is low.

**Table 2 entropy-23-01196-t002:** *SA* comparison of the six models.

	FCM	LINC	MICO	ARKFCM	LIC	WLSM
WM	0.8135	0.8551	0.9175	0.9518	0.9633	0.9787
GM	0.7632	0.8810	0.8453	0.9027	0.9174	0.9470
Average	0.7883	0.8680	0.8814	0.9272	0.9404	0.9628

**Table 3 entropy-23-01196-t003:** The iterations and calculative time of the four models.

	LINC	MICO	LIC	WLSM
Iterations	983	**54**	405	651
Time (s)	42.14	31.65	29.82	27.23

## Data Availability

Not applicable.

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
