# Peer review of "Magnetic Resonance Imaging Segmentation via Weighted Level Set Model Based on Local Kernel Metric and Spatial Constraint"

_entropy, 2021, doi:10.3390/e23091196_

Round 1
Reviewer 1 Report
The paper explores an interesting and challenging medical imaging problem; magnetic resonance imaging segmentation. Results have been reported on synthetic and real MRI images. Comments formulated during my review are presented below. These are as follows:
(1) The Authors should also avoke other recent papers dealing with MRI segmentation problem, namely:
(a) Integrating fuzzy entropy clustering with an improved PSO for MRI brain image segmentation. Applied Soft Computing, (2018), 65, 230-242.
(b) Semi–automatic corpus callosum segmentation and 3d visualization using active contour methods. Symmetry, 10(11), (2018), 589.
(c) Brain image segmentation based on FCM clustering algorithm and rough set. IEEE Access, (2019), 7, 12386-12396.
(2) In the related work section, a more rigorous investigation on the existing methods, such as comparison of previous approaches in terms of pros and cons, should be given. A summary table can be used in this regard.
(3) Figures with their captions should become self-explantory. What the blue and red colors mean? E.g. Figure 4 and 5, etc.
(4) Furthermore, where are the limitations of your study? Clarifying the limitations of a study allows the readers to understand better under which conditions the results should be interpreted.
(5) Future research work must be sufficiently widely argued. The authors need to open a real window for future work in the Conclusion section. The authors also need to clearly provide several solid future research directions in the Conclusion section.
Author Response
Response to Reviewer 1 Comments
Point 1: The Authors should also avoke other recent papers dealing with MRI segmentation problem, namely:
(a) Integrating fuzzy entropy clustering with an improved PSO for MRI brain image segmentation. Applied Soft Computing, (2018), 65, 230-242.
(b) Semi–automatic corpus callosum segmentation and 3d visualization using active contour methods. Symmetry, 10(11), (2018), 589.
(c) Brain image segmentation based on FCM clustering algorithm and rough set. IEEE Access, (2019), 7, 12386-12396.

Response 1: According to the reviewer's suggestions, three recommended literatures have been used as references in our manuscript, see Ref. [18], Ref. [25] and Ref. [26].
Point 2: In the related work section, a more rigorous investigation on the existing methods, such as comparison of previous approaches in terms of pros and cons, should be given. A summary table can be used in this regard.
Response 2: Considering the reviewer’s suggestion, we have given a summary Table 1 in the related work section.
Point 3: Figures with their captions should become self-explantory. What the blue and red colors mean? E.g. Figure 4 and 5, etc.
Response 3: Blue and red lines are the final zero-level contours of segmentation results, due to the multiple experimental results from Figure 5 to Figure 12 are involved, a unified explanation is given at the beginning of Section 3.
Point 4: Furthermore, where are the limitations of your study? Clarifying the limitations of a study allows the readers to understand better under which conditions the results should be interpreted.
Response 4: According to the reviewer's suggestions, we give the application conditions of the algorithm proposed in this paper, and point out the shortcomings and improvements of WLSM algorithm in section 4.
Point 5: Future research work must be sufficiently widely argued. The authors need to open a real window for future work in the Conclusion section. The authors also need to clearly provide several solid future research directions in the Conclusion section.
Response 5: According to the reviewer's suggestions, we have added corresponding content and provided several feasible future research directions in Conclusion section.

Reviewer 2 Report
The authors proposed a new method for improving the intensity normalization of MR images and its subsequent segmentation. The method was used on synthetic and real datasets with various dose levels and compared with other state-of-the-art (SOTA) methods. The detail given in the literature review and methods sections is appreciated, though too much for a research paper. I believe major restructuring of the paper needs to be done for publication:
- Sections 1 and 2 should be significantly shortened and combined. Only the methods relevant to the proposed and SOTA methods should be introduced. The SOTA algorithm can also be further summarized since previous works would have introduced it in more detail. Only the differences with the proposed method should be relevant here.
- There is not enough detail in Section 4. How many datasets and slices were used? How were they chosen? What is the demographic and diagnosis (if any) data?
- A Discussion section is missing. What are the significance and implications of this work?
- All the brains should have the anterior side facing up.
- Please have a professional English speaker thoroughly revise the manuscript.
Minor comments:
Line 33: Not necessarily true, as qualitative MRI reads are commonly used clinically. Please rephrase.
Line 457: Please define the ground truth. Also, would choosing other metrics such as the Dice coefficient influence the results?
Line 583: No need to repeat the results in the conclusion section.
Author Response
Response to Reviewer 2 Comments
The authors proposed a new method for improving the intensity normalization of MR images and its subsequent segmentation. The method was used on synthetic and real datasets with various dose levels and compared with other state-of-the-art (SOTA) methods. The detail given in the literature review and methods sections is appreciated, though too much for a research paper. I believe major restructuring of the paper needs to be done for publication:
Point 1:Sections 1 and 2 should be significantly shortened and combined. Only the methods relevant to the proposed and SOTA methods should be introduced. The SOTA algorithm can also be further summarized since previous works would have introduced it in more detail. Only the differences with the proposed method should be relevant here.
Response 1: According to the reviewer’s suggestions, Sections 1 and 2 have been combined, the content is simplified and only the parts closely related to this study are retained.
Point 2:There is not enough detail in Section 4. How many datasets and slices were used? How were they chosen? What is the demographic and diagnosis (if any) data?
Response 2: Based on the comments of the reviewer, we have supplemented the contents of section 4. The detailed information of the data set (BrainWeb and IBSR) used in the experiments is described at the beginning of section 4. At the same time, specific information such as the amount of data required in the experiment and the selected method are also provided in the corresponding subsections. The demographic and diagnosis data is not involved in the research content of this paper.
Point 3:A Discussion section is missing. What are the significance and implications of this work?
Response 3: We have added the discussion content in Section 1 and Conclusion section to illustrate the significance and purpose of this work.
Point 4:All the brains should have the anterior side facing up.
Response 4: According to the requirement of the research task, the sliced brain images in experiments from three planes (sagittal, coronal or axial), and these experimental data are displayed in a way that is convenient for observation and analysis.
Point 5:Please have a professional English speaker thoroughly revise the manuscript.
Response 5: According to the reviewer’s suggestions, we have polished and revised the full text of the manuscript by native English speaker.
Minor comments:
Point 1:Line 33: Not necessarily true, as qualitative MRI reads are commonly used clinically. Please rephrase.
Response 1: We have revised Line 33 according to the reviewer’s suggestions.
Point 2: Line 457: Please define the ground truth. Also, would choosing other metrics such as the Dice coefficient influence the results?
Response 2: Due to our negligence, there is a problem with the expression of segmentation accuracy (SA), we have given its new expression, thanks for the reviewer’s comments. In medical image segmentation, the main evaluation indicators of the proposed segmentation algorithm include segmentation accuracy (SA), Dice coefficient (DC), Jaccard similarity (JS), Hausdorff coefficient, etc. In order to avoid confusion between subjective and objective evaluation criteria, we select two typical evaluation indicators: SA and JS. Dice coefficient (DC) can also be used to evaluate the performance of the algorithms.
Point 3: Line 583: No need to repeat the results in the conclusion section.
Response 3: The redundant text has been deleted on Line 583

Reviewer 3 Report
In this work, the authors proposed a novel method per MRI segmentation based on level set, extending a previous method based on level set (LIC); thus, the novelty is limited. The manuscript needs an overall revision, and some sentences must be clarified. Some details about the equation in the related work section are missing or incoherent (I->J ?). I would change the title of section 2 since the related work are addressed in section 1.
The experimental section lacks details; in particular, the description of the used images/datasets is missing. Indeed, it is not clear if the used images are already corrupted by noise or if the authors added the noise (and how). The same goes for inhomogeneity.
Then the experiments should be performed with a certain schematicity; firstly, on images without noise to then gradually add some noise and inhomogeneity. Experiments should be consistent and always performed with the same metric, not firstly exploiting one index of similarity and then another. Since the table showing the quantitative results is only one, both metrics could be used (why not the DICE index? Is the most used to evaluate segmentation).
None segmentation method based on deep learning is mentioned, while a further comparison with this kind of methods should be introduced.
Some minor details
- line 401 Inter -> Intel
- line 467 nois -> noise
Author Response
Response to Reviewer 3 Comments
Point 1: In this work, the authors proposed a novel method per MRI segmentation based on level set, extending a previous method based on level set (LIC); thus, the novelty is limited. The manuscript needs an overall revision, and some sentences must be clarified. Some details about the equation in the related work section are missing or incoherent (I->J ?). I would change the title of section 2 since the related work are addressed in section 1.
Response 1: According to the reviewer’s suggestions, we have polished and revised the full text of the manuscript. In view of the redundancy of Section 2, we re-integrated Section 1 and Section 2 of this paper to make its content more concise.
Point 2: The experimental section lacks details; in particular, the description of the used images/datasets is missing. Indeed, it is not clear if the used images are already corrupted by noise or if the authors added the noise (and how). The same goes for inhomogeneity.
Response 2: Based on the comments of the reviewer, we have supplemented the contents of section 4. The detailed information of the data set (BrainWeb and IBSR) used in the experiments is described at the beginning of section 4. At the same time, specific information such as the amount of required data, noise level and intensity inhomogeneity are also provided in the corresponding subsections.
Point 3: Then the experiments should be performed with a certain schematicity; firstly, on images without noise to then gradually add some noise and inhomogeneity. Experiments should be consistent and always performed with the same metric, not firstly exploiting one index of similarity and then another. Since the table showing the quantitative results is only one, both metrics could be used (why not the DICE index? Is the most used to evaluate segmentation).
Response 3: The experimental objects in this manuscript are divided into two types, one is simulated brain MRI image, the other is real brain MRI image. The specific experimental steps: firstly, verify the robustness of the proposed algorithm to noise, secondly, evaluate the estimation ability of bias field, and thirdly, evaluate the segmentation ability of images corrupted by noise and bias field at the same time. In medical image segmentation, the main evaluation indicators of the proposed segmentation algorithm include segmentation accuracy (SA), Dice coefficient (DC), Jaccard similarity (JS), Hausdorff coefficient, etc. In order to avoid confusion between subjective and objective evaluation criteria, we select two typical evaluation indicators: SA and JS. Of course, Dice coefficient (DC) can also be used to evaluate the performance of the algorithms.
Point 4: None segmentation method based on deep learning is mentioned, while a further comparison with this kind of methods should be introduced.
Response 4: We have added a paragraph in Section 1 to mention the application of deep learning method in the field of medical image segmentation.
Point 5: Some minor details
- line 401 Inter -> Intel
- line 467 nois -> noise
Response 5: Thank you very much for the reminder of the reviewer, we have revised the corresponding problems.

Reviewer 4 Report
The manuscript proposes a weighted level set model (WLSM) to segment inhomogeneous intensity and noisy MRI images. Comparison with other level set based methods on both real and synthetic MRI images shows the advantage of WLSM.
Major comments:
- The authors compared the proposed method with several improved level set methods (LINC, MICO, and LIC). But there’s no comparison with any other types of segmentation methods.
- Please add more details for Chapter 4. Experiments and Results, e.g.,
-How many images were used for testing (both real and synthetic data sets)
-The size of those testing images
Minor comments:
- Please double check the format of all the figures and tables:
The subtitles of some figures are not centered (figure 3)
- In Figure 4, it’s better to add the segmentation results on the original images in Figure 4(a), so that it will be clearer to see the improvement after image enhancement.
- Please double check the grammar and typos, e.g.,
Line 37: which is aim at assign -> assigning
Line 228: Gaussian kernel distance in Figure 1(b) are greatly different -> is
Line 461: segmentation results -> results
Line 467: corrupted by 5% nois -> noise
Author Response
Response to Reviewer 4 Comments
The manuscript proposes a weighted level set model (WLSM) to segment inhomogeneous intensity and noisy MRI images. Comparison with other level set based methods on both real and synthetic MRI images shows the advantage of WLSM.
Major comments:
Point 1: The authors compared the proposed method with several improved level set methods (LINC, MICO, and LIC). But there’s no comparison with any other types of segmentation methods.
Response 1: Considering the reviewer’s suggestion, we have added two segmentation methods (FCM and ARKFCM ) to compare the segmentation performance in subsection 3.1, as shown in Table 3.
Point 2: Please add more details for Chapter 4. Experiments and Results, e.g.,
-How many images were used for testing (both real and synthetic data sets)
-The size of those testing images
Response 2: Thanks for the reviewer’s comments, and we have supplemented the contents of section 4. The detailed information of the data sets (BrainWeb and IBSR) used in the experiments is described at the beginning of section 4. At the same time, specific information such as the amount of data required in the experiment and the size of those testing images are also provided in the corresponding subsections.
Minor comments:
Point 1: Please double check the format of all the figures and tables:
The subtitles of some figures are not centered (figure 3)
Response 1: we have checked and revised the manuscript including text, figures and tables.
Point 2: In Figure 4, it’s better to add the segmentation results on the original images in Figure 4(a), so that it will be clearer to see the improvement after image enhancement.
Response 2: We added the ground truth (GT) as the comparison of segmentation results in Figure 4.
Point 3: Please double check the grammar and typos, e.g.,
Line 37: which is aim at assign -> assigning
Line 228: Gaussian kernel distance in Figure 1(b) are greatly different -> is
Line 461: segmentation results -> results
Line 467: corrupted by 5% nois -> noise
Response 3: Thank you very much for the reminder of the reviewer, and we have revised the corresponding problems.

Round 2
Reviewer 1 Report
The Authors have addressed all the comments.
Author Response
Response to Reviewer 1 Comments
Point : The Authors have addressed all the comments. 

Response : Thank you very much for your comments on our manuscript.

Reviewer 2 Report
The authors' efforts to improve the manuscript are appreciated. However, points 3 and 4 were not addressed. According to Entropy rules there should be a standalone discussion for research manuscripts, normally between the results and conclusions sections. Also, for axial images the anterior side should face up, since that is how radiologists view the images.
Author Response
Response to Reviewer 2 Comments
Point 1: The authors' efforts to improve the manuscript are appreciated. However, points 3 and 4 were not addressed. According to Entropy rules there should be a standalone discussion for research manuscripts, normally between the results and conclusions sections.
Response 1: Considering the reviewer's comments, we have added a standalone discussion section before section 5.
Point 2: Also, for axial images the anterior side should face up, since that is how radiologists view the images.
Response 2: According to the reviewer’s suggestion, we have adjusted the tested axial images, and the anterior sides of all corresponding images are faced up.

Reviewer 3 Report
The authors revised taking into account all the referee's comments, thus it can be accepted for publication
Author Response
Response to Reviewer 3 Comments
Point: The authors revised taking into account all the referee's comments, thus it can be accepted for publication 

Response: Thank you very much for your comments on our manuscript.

Reviewer 4 Report
The manuscript proposes a weighted level set model (WLSM) to segment inhomogeneous intensity and noisy MRI images. Comparison with other level set and fuzzy C-means based methods on both real and synthetic MRI images shows the advantage of WLSM.
Major comments:
- There’s no explanation why FCM and ARKFCM were taken into consideration for performance comparison. And the results of FCM and ARKFCM were not shown in JSC comparison part.
- The authors should add the results of FCM and ARKFCM in Figure 6 and 9-12.
Minor comments:
- Please double check the grammar and typos again, e.g.,
Line 106: segementation performance -> segmentation performance
Author Response
Response to Reviewer 4 Comments
Major comments:
Point 1: There’s no explanation why FCM and ARKFCM were taken into consideration for performance comparison. And the results of FCM and ARKFCM were not shown in JSC comparison part.
Response 1: Considering the reviewer's comments, we have added the explanation why FCM and ARKFCM were taken into consideration for performance comparison in subsection 3.1. In subsection 3.2, the purpose of our experiments is to test bias field correction ability of the proposed algorithm, because FCM and ARKFCM algorithms cannot estimate the bias field in brain MRI images, we only compared LINC, MICO, LIC and WLSM algorithms with bias field correction ability in the experiments of subsection3.2.
Point 2: The authors should add the results of FCM and ARKFCM in Figure 6 and 9-12.
Response 2: According to the reviewer’s suggestion, we have added the experimental results of FCM and ARKFCM in Figures 6, 11 and 12, respectively. In Figures 9 and 10, we mainly compared the bias field correction ability of the algorithms, so the experimental results of FCM and ARKFCM are not shown.
Minor comments:
Point 1: Please double check the grammar and typos again, e.g.,
Response 1: According to the reviewer’s suggestions, we have checked the grammar and typos of the manuscript.
Point 2: Line 106: segementation performance -> segmentation performance
Response 2: Thank you very much for the reminder of the reviewer, and we have revised the corresponding problems.

Round 3
Reviewer 2 Report
Before publication, in continuation of the previous comments, I would recommend that the last paragraph in the conclusions section be moved to the discussion. Also, the brains in figure 2 also need to be rotated to anterior-side-up.
Author Response
Response to Reviewer 2 Comments
Point: Before publication, in continuation of the previous comments, I would recommend that the last paragraph in the conclusions section be moved to the discussion. Also, the brains in figure 2 also need to be rotated to anterior-side-up.
Response: Considering the reviewer's comments, we have moved the last paragraph in the conclusions section to the discussion section, and the anterior sides of the brains in figure 2 are faced up. Thank you very much for the suggestion of the reviewer.

Reviewer 4 Report
The manuscript proposes a weighted level set model (WLSM) to segment inhomogeneous intensity and noisy MRI images. Comparison with other level set and fuzzy C-means based methods on both real and synthetic MRI images shows the advantage of WLSM.
Author Response
Response to Reviewer 4 Comments
Point : The manuscript proposes a weighted level set model (WLSM) to segment inhomogeneous intensity and noisy MRI images. Comparison with other level set and fuzzy C-means based methods on both real and synthetic MRI images shows the advantage of WLSM. 

Response : Thank you very much for your comments on our manuscript.
